# Did Covid-19 lockdown positively affect the urban environment and UN- Sustainable Development Goals?

**Ritwik Nigam[1], Gaurav Tripathi[2], Tannu Priya[2], Alvarinho J. Luis[3], Eric Vaz[4], Shashikant Kumar[5], Achala Shakya[6], Bruno Damásio[7]\*, Mahender Kotha[1]**

**1** School of Earth, Ocean and Atmospheric Sciences (SEOAS), Goa University, Taleigao, Goa, India, **2** Department of Geoinformatics, Central University of Jharkhand, Ranchi, Jharkhand, India, **3** Polar Remote Sensing Section, National Centre of Polar and Ocean Research, Ministry of Earth Science, Govt. of India, Headland Sada, Goa, India, **4** Department of Geography and Environmental Studies, Ryerson University, Toronto, Ontario, Canada, **5** Department of Architecture, Parul University, Limda, Gujarat, India, **6** Department of Computer Engineering, University of Petroleum and Energy Studies, Derhradun, India, **7** NOVA Information Management School (NOVA IMS), Universidade Nova de Lisboa, Campus de Campolide, Lisboa, Portugal

\* bdamasio@novaims.unl.pt

**Data Availability Statement:** The data are available at Kaggle, through the following link: https://www.kaggle.com/datasets/bdamasio/alphacity.

## Abstract

This work quantifies the impact of pre-, during- and post-lockdown periods of 2020 and 2019 imposed due to COVID-19, with regards to a set of satellite-based environmental parameters (greenness using Normalized Difference Vegetation and water indices, land surface temperature, night-time light, and energy consumption) in five alpha cities (Kuala Lumpur, Mexico, greater Mumbai, Sao Paulo, Toronto). We have inferenced our results with an extensive questionnaire-based survey of expert opinions about the environment-related UN Sustainable Development Goals (SDGs). Results showed considerable variation due to the lockdown on environment-related SDGs. The growth in the urban environmental variables during lockdown phase 2020 relative to a similar period in 2019 varied from 13.92% for Toronto to 13.76% for greater Mumbai to 21.55% for Kuala Lumpur; it dropped to −10.56% for Mexico and −1.23% for Sao Paulo city. The total lockdown was more effective in revitalizing the urban environment than partial lockdown. Our results also indicated that Greater Mumbai and Toronto, which were under a total lockdown, had observed positive influence on cumulative urban environment. While in other cities (Mexico City, Sao Paulo) where partial lockdown was implemented, cumulative lockdown effects were found to be in deficit for a similar period in 2019, mainly due to partial restrictions on transportation and shopping activities. The only exception was Kuala Lumpur which observed surplus growth while having partial lockdown because the restrictions were only partial during the festival of Ramadan. Cumulatively, COVID-19 lockdown has contributed significantly towards actions to reduce degradation of natural habitat (fulfilling SDG-15, target 15.5), increment in available water content in Sao Paulo urban area(SDG-6, target 6.6), reduction in NTL resulting in reduicied per capita energy consumption (SDG–13, target 13.3).

**Funding:** The author(s) received no specific funding for this work.

**Competing interests:** The authors have declared that no competing interests exist.

## Introduction

The Coronavirus disease that causes COVID-19 has global scale transmission. Globally, it caused over 5.975 million deaths (as on March 1, 2022) and led to a contraction in the economy due to disruption of production and supply market chain, and caused public health emergencies. Many countries restorted to impose partial to total lockdown measures [1] to arrest the COVID-19 transmission. Containment measures such as wearing a mask, social distancing, quarantining, screening tests were among the most significant challenges witnessed in the modern era.

Several studies have highlighted the positive impact of COVID-19 lockdown on the environment [2, 3]. A case study on the electricity consumption in China revealed that the electricity usage was on average 29% lower than that in the pandemic-free scenario [4]. Furthermore, China's carbon emissions during 2020 dropped by 8.8%, while that for India, the US and the EU it was a higher drop of 19.2%, 16.6% and 19.8%, respectively [5]. While all the of above were well documented because COVID-19 pandemic has greatly accelerated research on the integration of digital technologies (big data, artificial intelligence, cloud computing, 5G) to combat the COVID-19 pandemic and healthcare [6].

The post-COVID spillover effects of China's economic growth (in terms of economy and energy) have the most obvious impact on upper-middle-income countries' economic growth (0.17%), followed by the economic growth of lower-middle-income countries (0.16%) and high-income countries (0.15%) [7]. However, the spillover effect of China's economic growth has the most significant impact on energy consumption in high-income countries (0.11–0.45%), followed by energy consumption in upper-middle-income countries (0.08–0.33%) and in lower-middle-income countries (-0.02–0.05%) [8].

A drastic reduction in $NO_2$ across China due to a slowdown in economic activities contributed to a decrease in air pollution [9]. A substantial drop in air pollutants in 44 Chinese cities was attributed to a 69.85% reduction in human mobility [10]. Considerable air quality improvement was observed during the lockdown when compared to pre- and post-lockdown in India [11]. Air pollution in Wuhan, Italy, the US, and Spain had reduced by 30%, mainly due to a 90% drop in human mobility during lockdown [2]. Lockdown propelled a 17% increase in ozone and a 56% reduction of $NO_x$ in Italian cities [12]. The electricity demand in Ontario was reduced by 14% and $CO_2$ emission dropped by 4000 tons during April 2020 lockdown [13]. Reduction in traffic and industrial activity dwindled brought down the energy consumption during the lockdown in four metropolitan cities of India, which improved the air quality [14].

Rapid urbanization has led to haphazard development, wherein the economy and environment are inversely proportional to each other. Large economic centers like New York–also an 'Alpha+' City–have turned into a concrete jungle [15] that needs a bulk of energy to sustain. An alpha city is a city that is a primary node in the global economic network [16]. These large metro cities have grown significantly, over-exploiting their natural environment in pursuit of economic development. Conversely, these cities are responsible for sustaining the national economy and dealing with intense pressure and competitiveness among global cities, leaving little space for a sustainable environment and energy. The restriction of economic activities, transportation, industries during lockdown provided more or less a period of undisturbed growth to urban vegetation and low energy consumption [13].

The United Nations (UN) General Assembly has declared the current decade as the UN decade on ecosystem restoration to prevent, halt and reverse the degradation of ecosystems on every continent. The anthropogenic activities, regardless of their intensity, impact the natural environment in different ways. In an attempt to move forward, the present study is aimed to

investigate the role of lockdown in influencing urban natural settings such as urban greenness, energy consumption, and land surface temperature for five Alpha cities by comparing the 2020 lockdown period with the similar period of 2019 and pre- and post- lockdown phases of 2020. The availability of data, economic background, and their representativeness in different continents were some characteristics for selecting the alpha cities (Kuala Lumpur, Mexico, greater Mumbai, Sao Paulo, and Toronto). This study is a first attempt in exploring the positive effects of COVID-19 lockdown restrictions on environment–related UN Sustainable Development Goals (UN–SDG) by synthesizing the current results and leading environmental expert perceptions towards it.

## Methods

The present study attempted to determine and find evidence of the role of COVID-19 induced lockdown restrictions in variation in the urban natural environment in totality. Therefore, the study used different sets of remote sensing data (S1 Table in S1 File) as the proxies to complement each of the urban natural environment variables assessed in the present study. The steps involved in the present study are illustrated in S1 Fig in S1 File.

### Satellite data acquisition via Google Earth Engine platform

Various functions were used in the first step to access or acquire a satellite dataset by calling a stack or series of images from the repository provided by Google Earth Engine platform (GEE) itself and made a composite for all required lockdown phases, cities, and years. Each of them has its Image Collection and ID in GEE. In most of the cases, Landsat OLI based optical satellite data were available with more than 50% cloud cover, which led us to use MODIS based LST product (MOD11A2) at a spatial resolution of 1 km and Visible Infrared Imaging Radiometer Suite Night-time light data (VIIRS NTL data) at a spatial resolution of 15 arc second. Individual images and images merged from an existing database were analyzed using GEE.

A filter function was scripted to limit the acquired image to only in its chosen location and date of interest. As the images suffered from cloud cover, a function to mask out the clouds using a pixel QA cloud band value provided within the products was utilized.

### Satellite data derived urban natural environment variables

The steps involved in the methodology for calculating urban natural environment variables/ proxies are given below.

**Normalized Difference Vegetation Index.** The Normalized Difference Vegetation Index (NDVI) was calculated in this study using Sentinel 2A/B data at a spatial resolution of 10×10m and 15-day interval. NDVI was calculated for pre-, during, and post-lockdown phases as per the table for 2019 and 2020 to detect changes in surface vegetation [17]. NDVI for each particular phase was calculated using near-infrared (NIR) and red (RED band) using the following equation:

$$NDVI = \frac{NIR - RED}{NIR + RED} \tag{1}$$

**Normalized Difference Water Index.** The Normalized Difference Water Index (NDWI) was proposed in 1996 to detect surface waters in wetland environments and measure surface water extent [18]. Although the index was devised for its use with Sentinel–2A/ B data, it has been successfully used with other sensor systems in applications where the measurement of the extent of open water is needed [19] NDWI is computed using the near-infrared (NIR–

SENTINEL-2, band 8) and the short-wave infrared (SWIR–SENTINEL-2, band 12) reflectance. NDWI was calculated using:

$$NDWI = \frac{Green - NIR}{Green + NIR} \tag{2}$$

Ref. [31] asserted that values of NDWI greater than zero are assumed to represent water surfaces, while values less than or equal to zero are assumed to be non-water surfaces [19].

**Land Surface Temperature.**   Land Surface Temperature (LST) is one of the most prominent parameters in urban climatological studies that characterize the temperature arising out of different land-use [32]. Remote sensing-derived thermal imageries are used to determine thermal characteristics of land surface [20]. The LST can provide information related to fluctuation in surface temperature due to various determinants such as concretization, massive alterations in land-use cover (vegetation to built-up), increment in population density, vehicular movement, etc. disturbs the surrounding natural environment. In the USA, it was found that due to urbanization, there is an increase in temperature by 1˚C per 100,000 populates [21]. Spatial variation in LST was observed in Delhi due to differences in population density [20], which indicates that high population density leads to higher population movement at the local level, leading to an increase in the LST. It can be used as a proxy indicator to determine variations in LST during the lockdown period due to population movement restrictions. For the LST MODIS MOD11A2 product, the LST_DAY_1km band was used [22 Wan, Zhengming, Hook, Simon & Hulley, Glynn]. The temperature is expressed in Celsius.

**Night Time Light Data.**   Night-time lights (NTL) is generated due to illumination of various earth features like urban infrastructure such roads, buildings, airports, etc. and over-illumination of advertisement hoardings in the urban areas leads to light pollution [22–26], as well as, in smaller proportion, due to natural phenomenon like Aurora, lightening, active volcano and natural hazards also emit NTL [27 (NASA)]. Therefore, it is increasingly used as a proxy indicator of anthropogenic energy consumption and emission [28] and for detecting light pollution mainly in urban areas [29]. The NTL parameter was used in the present study to determine the spatio-temporal variations in the NTL (to determine variations in energy consumption) during the lockdown and post-lockdown periods. We used monthly average radiance composite images using night-time data from the Visible Infrared Imaging Radiometer Suite (VIIRS) Day/Night Band (DNB).

**Difference calculation of NDVI, NDWI, LST, and NTL.**   In this work, indices (NDVI and NDWI) and satellite-based products (LST and NTL) were calculated for 2020 and 2019, with the latter as the base data. The impact of pre-, post-, and during lockdown phases was evaluated using various image statistics such as image difference, percentage image difference, image correlation, peak signal-noise ratio (P-SNR), and structural similarity measure (SSIM).

First, image difference was calculated using the absolute subtraction method. Second, the percentage difference was calculated by considering images of 2019 as a reference image to determine the extent of changes in two different periods. Later on, the P-SNR values were also computed, which shows the peak signal-to-noise ratio, in decibels, between two images [30]. This ratio was further used as a quality measurement between the original and a compressed image. The higher the PSNR, the better the quality of the compressed or reconstructed image [30, 31].

Image correlation was also derived by computing the correlation coefficient between the images processed with a median filter. Finally, structural Similarity measures (SSIM) were calculated to evaluate the structural similarity for a grayscale image using the reference image [29].

Finally, the mean difference in percentage was calculated using:

$$\frac{\text{New value} - \text{Old value}}{\text{Old value}} \times 100 \qquad (3)$$

where "New value" refers to the mean value for 2020 and the "Old value" refers to the mean value for 2019 of each lockdown period for each city (see the next section).

**Technical characteristics of the processed data.**   The measures such as pixel difference, peak signal-to-noise ratio, SSIM, correlation coefficient and percentage mean difference between 2019 and the similar period in 2020 are provided in S3 Table in S1 File. This attributes are important to highlight the quality of the data output. We discuss these for each of the alpha cities in the following sections.

**Linkage of lockdown impact on environment-related SDGs.**   A novel methodology is formulated to explore and analyze the linkages. We used the Thurstone scale to formulate an online questionnaire. The scale is a psychological scale that is used to investigate and quantify expert responses about the conceptual subjects to compare them with each other and come up with some quantifiable solutions.

1. Reconnaissance survey of UN-SDG literature: Out of a total of 17 SDGs and their 169 targets, 25 targets from 10 different SDGs somehow related to the urban environment were selected to explore the possibilities of linkage.

2. All the targets were converted into statements.

3. To determine a median score of each statement, a panel of judges comprising field experts scored each of the statements on a scale of 1 to 11. Higher the score, the stronger the statement supporting the subject matter.

4. A questionnaire was constructed using these statements containing responses such as "Agree", "Disagree", and "Can't say".

5. An online questionnaire survey was conducted.

6. The percentage share of each response concerning each statement was calculated to quantify the statements.

7. Finally, the percentage share values were plotted.

8. NDVI, NDWI, LST, and NTL results during the lockdown phase of five alpha cities were used to validate questionnaire responses to prove the linkage.

**Quantification of the impact of lockdown on Alpha cities.**   COVID-19 induced lockdowns (total and partial) had different sets of restrictions, implying that both had influenced the urban natural environment differently. The quantification would assist in determining which of them had a more positive impact on Alpha cities' urban natural environment. Details are available in the supplementary section.

The addition of NDVI and NDWI mean difference (in %) during the lockdown and the subtraction of NTL and LST mean difference (in %) values during lockdown divided by the total number of urban environmental variables used in the study.

$$\frac{(\text{NDVI} + \text{NDWI}) + (-\text{LST}) + (-\text{NTL})}{\text{Total number of environmental variables}} \qquad (4)$$

(-LST) and (-NTL) applied, in case they are positive otherwise, we used,

$$\frac{(NDVI + NDWI) + (LST) + (NTL)}{Total\ number\ of\ environmental\ variables} \tag{5}$$

The LST and NTL are inversely proportional to positive urban environmental health. Therefore, to determine positive growth, their mean difference (in %) values, if positive, were subtracted; otherwise, they were added to NDVI, and NDWI mean difference (in %) during the lockdown phase. For example, Sao Paulo = NDVI (9.67) + NDWI (3.57) + LST (1.67) + NTL (-19.84)/4. Here, LST was already in negative, so it was added while NTL was subtracted as it was surplus.

## Study area- Alpha cities

The concept of Alpha city emerges from geography and urban studies the momenclature of which is given by the Globalization and World Cities and Research Network (GaWC) [19] founded in 1998 [32]. These are large cities around the world which are ranked based on their context of globalization and connectivity. The ranking is based upon different parameters in which economic parameter is heavily weighted against socio-cultural parameters. The economic parameter is based on "the magnitude of business service connection of the city to 707 other major cities" [33]. Based on the "international connectedness," the think-tank classifies potential cities into Alpha, Beta, and Gamma tiers [34]. Alpha cities are directly linked with major world economic cities and are the major economic powerhouse of their respective countries. They are classified as Alpha++, Alpha+, Alpha, Alpha- cities. Beta cities are moderately connected to world economic cities, and they also serve as an economic powerhouse of their home countries. They are Beta+. Beta and Beta− cities. Gamma cities are linked partially on the smaller scale to the world economic cities; they are Gamma+, Gamma, and Gamma- cities.

The present study included five Alpha cities: two Asian (Kuala-Lumpur and greater Mumbai), two from North America (Mexico City and Toronto), and one from South America (Sao Paulo) S2 Fig in S1 File, general characteristics of the cities chosen in the study are summarized in S2 Table in S1 File.

## Results and discussion

Salient features of results are summarized of city-wise variation in various urban environmental parameters and the effect of lockdown and same is shown in Table 1 and city-wise spatio-temporal changes in NDVI, NDWI, LST and NTL based on the thematic maps of the five Alpha cities is illustrated in Figs 1 to 5. The specific changes observed during lockdown and post-lockdown are discussed city-wise.

### Kuala Lumpur city (Fig 1)

With the implementation of a partial lockdown on March 18, 2020 in Kuala Lumpur, the urban vegetation during the dry season (March-April) have increased substantially (Fig 2), despite the low amount of rainfall (mean: 5.84 mm) during the 2020 lockdown compared to a similar period in 2019 (mean: 7.07 mm) [35]. No change in water content (compared to 2019) occurred during March-April (Fig 2, and S1 Table in S1 File) due to low rainfall. Moreover, higher water content during pre- (February) and post-lockdown (June-July) lowered the LST compared to the March-April period. Weather conditions influenced LST during different lockdown phases (Table 1, Fig 1, and S1 Table in S1 File), and a result of anthropogenic activities coal burning, highway construction, increase in built-up areas between 1990–2015 [36] and airflow towards urban neighborhoods of Kuala Lumpur [37]. So, the LST variation results

**Table 1. Summary of variations in urban environmental parameters and the effect of lockdown.**

| | Variable | Kuala Lumpur | Mexico | Greater Mumbai | Sao Paulo | Toronto |
|---|---|---|---|---|---|---|
| NDVI | PD | 2% drop is seen between pre- and post-lockdown | remained constant at 56%. | Did not reflect any change from pre- to post-lockdown | marginal (1%) increase from post- to lockdown | 1% increase for the post-lockdown phase |
| | P_SNR | changed marginally | dropped from 20 to 17.6 from pre- to post-lockdown | dropped from 23 to 14.2, from lockdown to post-lockdown phase | doubled from pre- to post-lockdown | 17% increase from pre- to lockdown and 20% post-lockdown decrease |
| | SSIM | found to be greater than 59% | constant at 0.82 | dropped from 0.92 to 0.76, from the lockdown to post-lockdown | increased by 21% and 18% from pre- to lockdown and thereafter | decreased by 25% (5%) during lockdown (post-lockdown). |
| | R | exceeded 0.68 | exceeded 0.9 | dropped from 0.97 to 0.81 from lockdown to post-lockdown | increasing trend from pre- to lockdown | highest correlation (0.64) was found during the lockdown phase |
| | % MD | increased from 16% to +77% (pre-lockdown to lockdown); 118.17% drop (post-lockdown) | −20.83% in pre-lockdown reduced to −22.72% in lockdown; increased to 10.20% post-lockdown | −41.17% (pre-lockdown); reduced to zero during lockdown; increased to 21.87% post-lockdown | −20.83% in pre-lockdown; increased to 9.67% during lockdown and dropped to 5.71% post lockdown | whooping −500% during pre-lockdown; reduced to −18.18% during lockdown & dropped to −154.54% post-lockdown |
| NDWI | PD | dropped from 68% to 65% (pre- to lockdown); marginal increase to 66% (post-lockdown) | showed an increasing trend from 47 to 55% | 4% increase | 5% drop in lockdown compared to pre-lockdown; further 27% decrease in post-lockdown | found to be invariable (49%) |
| | P_SNR | greater than 10 | dropped from 20.9 to 13 | dropped by 23% and 12% from the pre- to lockdown | dropped by 5% during the lockdown | Increase of 53% from pre-lockdown to lockdown |
| | SSIM | exceeded 50%. Higher SSIM observed during and post-lockdown | reduced from 0.79 to 0.58 from pre- to post-lockdown phase | reduced by 11% (pre-, to lockdown); further 4% drop (lockdown to post-lockdown) | dropped by 7% during the lockdown, but increased by 78% during post-lockdown | |
| | R | increased from 0.6 to >0.7 during and post-lockdown | decreased from 0.94 to 0.76, corresponding to pre-, & post-lockdown | dropped by 0.14 (pre-lockdown period); increase by 4% from lockdown to the post-lockdown | drop during the lockdown by 5%, followed by an increase (0.99) during post-lockdown phase | increased from 0.5 to 0.73 from pre- to post-lockdown |
| | % MD | 15.38% pre-lockdown to nil during lockdown; increased to 14.28% post lockdown | 12.5% (pre-lockdown) to 3.7% (during lockdown), but increased to 14.7% in the post-lockdown | 31.25% for pre-lockdown; increased to 41.66% (lockdown); reduced to nil (post-lockdown | −3.12% pre-lockdown; increased to 3.57% in lockdown & slight drop to −3.57% post-lockdown | −16.66% pre-lockdown which increased to −7.40% in lockdown and dropped to 3.70% post-lockdown |
| LST | PD | not shown any changes during the three phases | observed to be zero | dropped by 35% from pre- to post-lockdown phase | | 38% drop in lockdown & 99% post-lockdown |
| | P_SNR | | | 20 to 24.5 increase from pre- to post-lockdown | | gradually increase from 5 to 34 from pre-, to post-lockdown |
| | SSIM | | during all three phases were a unity | gradual increase (0.93 to 0.98) from pre-, to post-lockdown | | increase of 93% in lockdown & 18% post-lockdown |
| | R | | unity in all three phases | very high (>0.9). | | high (> 0.7) in all 3 phases |
| | % MD | −12.64% (pre-lockdown), 12.12% increase(lockdown) & −9.60% post lockdown | 11.99% (pre-lockdown); increased to 16.85% during lockdown; reduced to 1.09% post-lockdown | −22.44% for pre-lockdown; dropped to −2.86% in lockdown & to 0.26% post lockdown | decreased from −0.23% (pre-lockdown) to −1.67% and −2.50% in lockdown and post-lockdown | −298.72% during pre-lockdown, 16.90% during the lockdown and 11.73% post- lockdown |

(*Continued*)

**Table 1.** (Continued)

|  | Variable | Kuala Lumpur | Mexico | Greater Mumbai | Sao Paulo | Toronto |
|---|---|---|---|---|---|---|
| NTL | PD | showed 17% during the post-lockdown phase | 16.1% to 15.2% (pre-, lockdown) to 18.6% in post-lockdown | increased from nil in lockdown to 0.59% post-lockdown | increased by 3% and 10% during the lockdown and post-lockdown | no variation for the three phases |
|  | P_ SNR |  | Drop (27.5 to 20.4) in pre-, to lockdown; 26% post-lockdown |  | increased by 7% during post-lockdown compared to lockdown |  |
|  | SSIM | >0.8 in all three phases | 1% drop from pre- to lockdown |  | exceeded 0.9 for all phases |  |
|  | R | >0.6 during all phases | exceeded 0.98 for all the phases |  |  |  |
|  | % MD | −9.60% for pre- lockdown; marginally declined to −9.64% in lockdown; reduced to −20.66% post lockdown | increased subsequently during each phase from 1.19% (pre-lockdown) to 7.81% (post-lockdown). | Reduced from −11.86% (pre-lockdown) to −10.55% (during lockdown) and 51.91% (post-lockdown) | −3.33% (pre-lockdown) and 19.84% (during the lockdown), which plunged to -44.55% (post-lockdown) | from −57.73% (pre-lockdown) dropped to −6.97% (during lockdown) and increased to 9.14% (post-lockdown). |
| **Effect of Lockdown** |  | surplus (21.55%) relative to similar period 2019 | deficit (−10.56%) relative to similar period in 2019 | positive 13.76%, compared to a similar period in 2019 | deficit −1.23% compared to a similar period in 2019 | positive 13.92% compared to similar period in 2019 |

PD- Pixel Difference; P_SNR- peak signal-noise ratio; SSIM- structural similarity measure; R- Correlation Coefficient; MD-Mean Difference

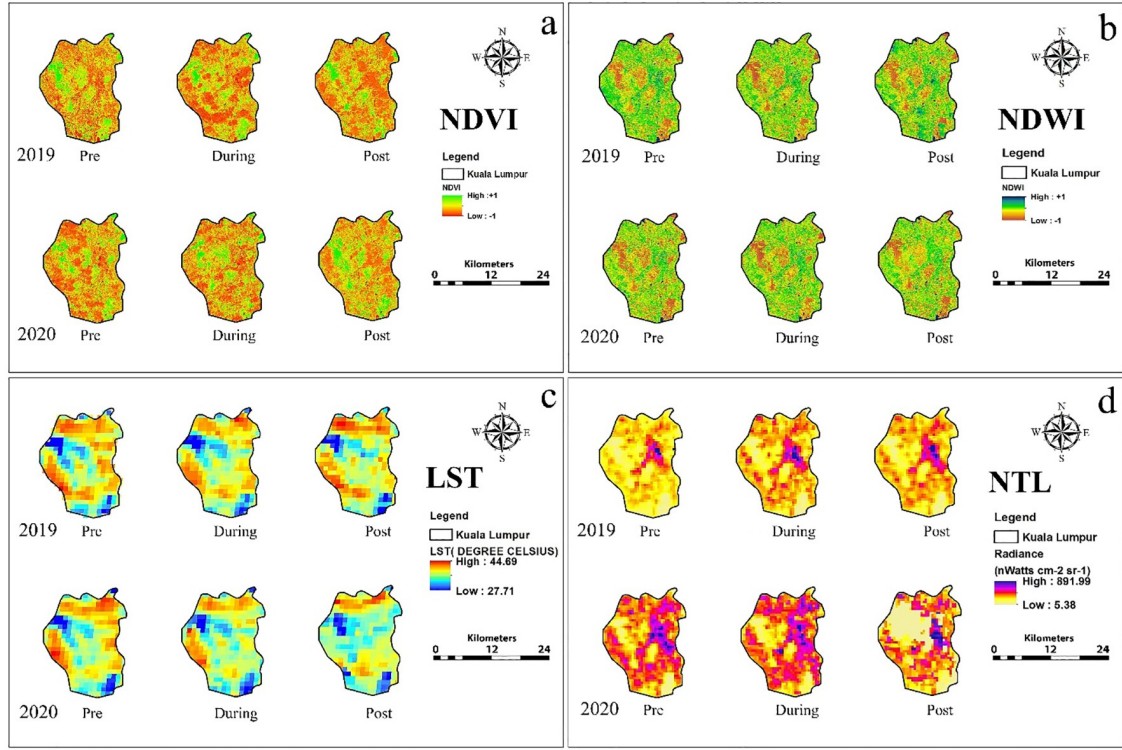

**Fig 1. Spatio-temporal changes in a) NDVI, b) NDWI, c) LST and d) NTL based on the thematic maps of the Kuala Lumpur city.**

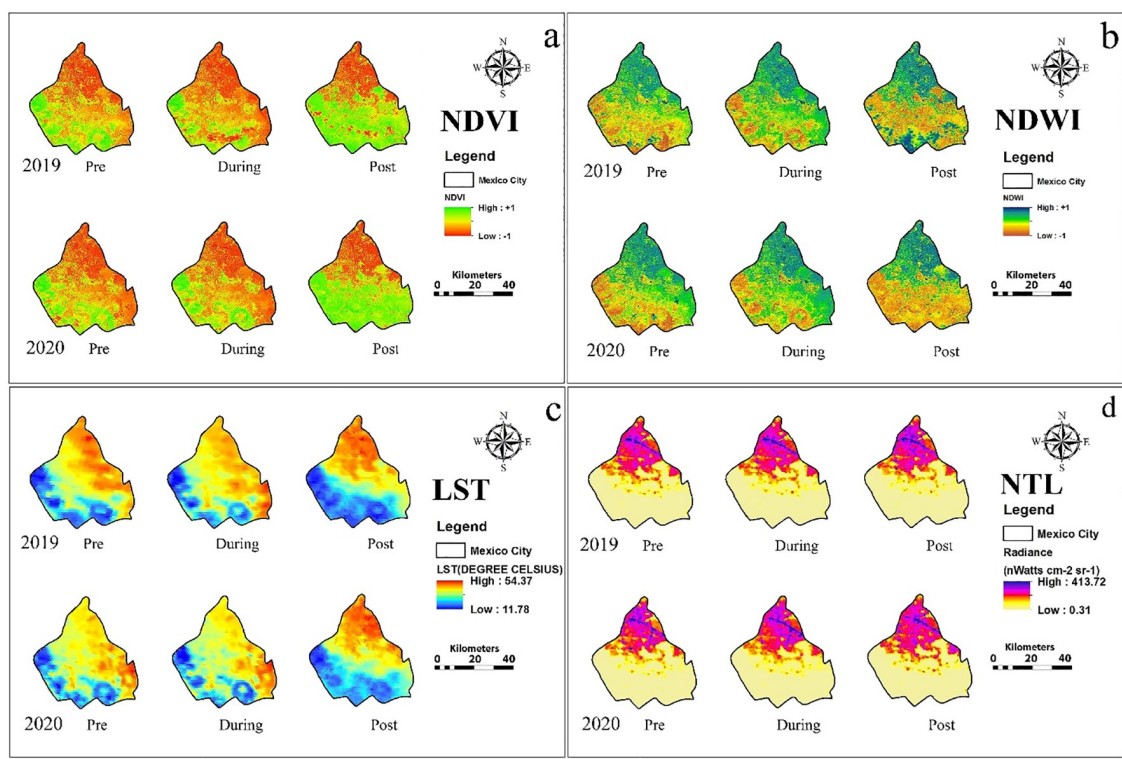

**Fig 2. Spatio-temporal changes in a) NDVI, b) NDWI, c) LST and d) NTL based on the thematic maps of the Mexico city.**

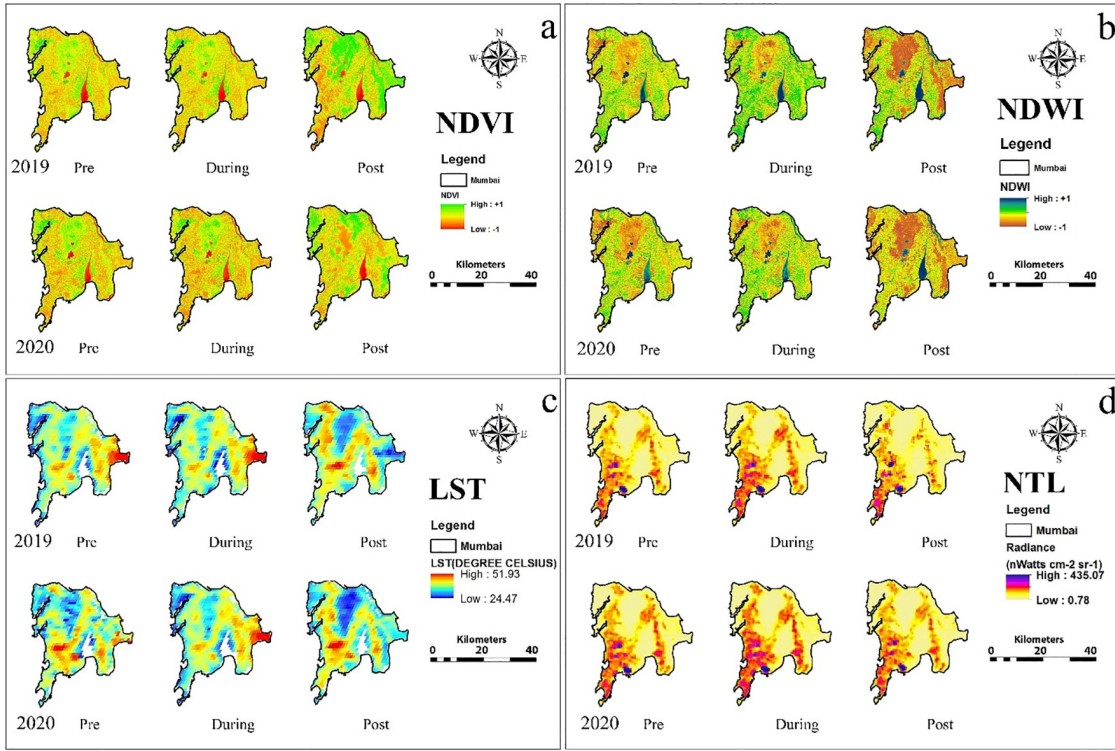

**Fig 3. Spatio-temporal changes in a) NDVI, b) NDWI, c) LST and d) NTL based on the thematic maps of the Mumbai city.**

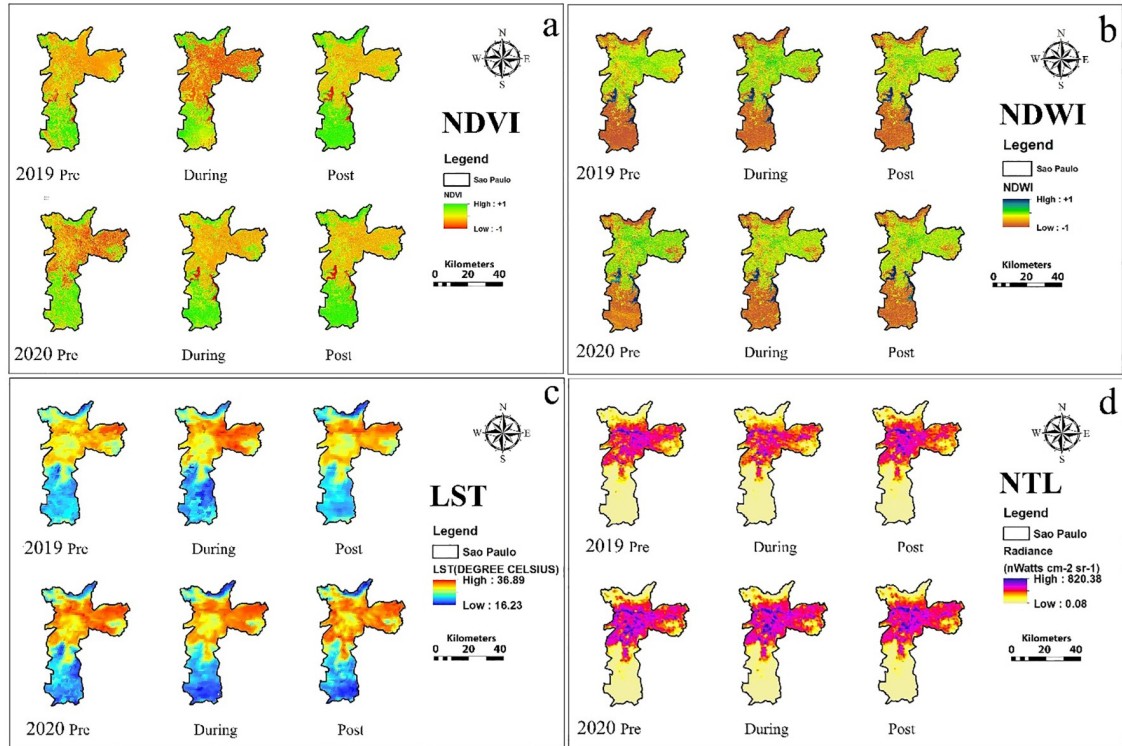

**Fig 4. Spatio-temporal changes in a) NDVI, b) NDWI, c) LST and d) NTL based on the thematic maps of the Sao Paulo city.**

from the continuous changes in land use patterns (mainly built-up) between 2019–2020, which is also evident from the higher pre- and post-lockdown phases. Thus, no LST change during the lockdown phase suggests restrictions on vehicle mobility. During the three phases of 2020, the active city lights showed a significant reduction in the prominent entertainment/ shopping area of Bukit Bintang compared to 2019 (Fig 1, S1 Table in S1 File); however, higher night-time illumination was found in northern, western, and southern regions of the city (Fig 1).

## Mexico city (Fig 2)

Mexico City implemented a partial lockdown on March 30, 2020. Precipitation was negligible during pre-lockdown (0.11 & 0.33 mm) and lockdown (0.82 & 0.14 mm) phases of 2020 and 2019 [38]. Post lockdown (August) rainfall was just 5.63 mm (2020) and 5.09 mm (2019), so the influence of precipitation on the urban environment of Mexico City during the three phases is minimal.

During the lockdown there was a reduction in the urban vegetation growth (S2 Fig in S1 File, S1 Table in S1 File) mainly due to the exceptionally higher temperature in 2020 (14˚– 30˚C) compared to (12˚–18˚C) in 2019 [39, 40]. The higher temperatures in 2019 compared to 2020 led to vegetation growth during pre-lockdown (11˚–20˚C in 2019 and 14˚–17˚C in 2020) and post-lockdown (16˚–19˚C in 2019 and 16˚–17˚C in 2020). Similarly, air temperature changed the water content across the lockdown phases by influencing the evaporation rate, which led to an increase in NDWI in 2020 due to lower air temperature compared to 2019 (Table 1, Fig 2, and S1 Table in S1 File). However, this phenomenon was negligible during the

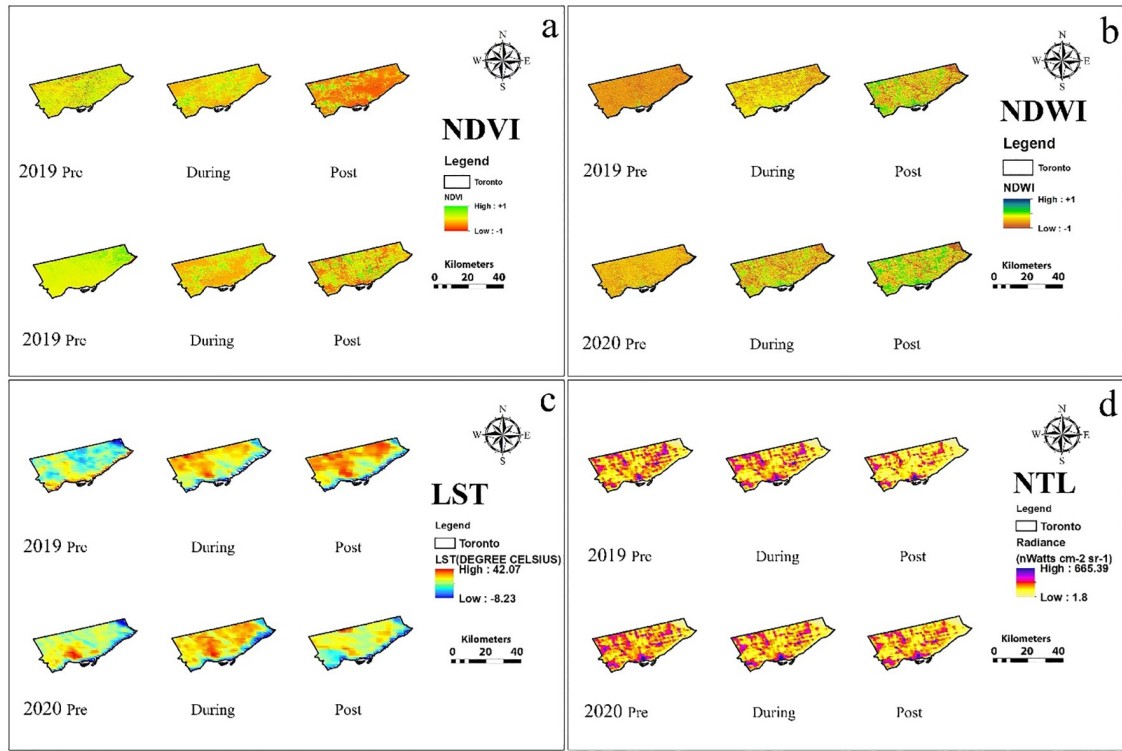

**Fig 5. Spatio-temporal changes in a) NDVI, b) NDWI, c) LST and d) NTL based on the thematic maps of the Toronto city.**

pre- and lockdown phases because there was an increase in household water consumption during lockdown [41, 42].

Compared to 2019, the LST was higher during the lockdown phase due to an increase of 10°C in the maximum daily temperature in 2020 (Fig 2, S3 Table in S1 File). Moreover, higher LST during the lockdown phase was one of the causes for the reduction of NDVI. During all the phases, the city observed the highest amount of change in the mean difference of NTL among all the alpha cities (Fig 2, S3 Table in S1 File). We found positive NTL changes between during and post-lockdown periods, while pre-lockdown had a negligible change (1.19%). From Industrial Vallejo (north-west) to central heritage area and highway 150D (east-central), the highest reduction in active lights was observed during and post-lockdown due to less vehicle mobility. However, south-central areas of Coyoacan, Coapa, Granjas Coapa, Toerillo Guerra, Jardines Del Pedregal, etc., revealed a higher illumination during and post-lockdown compared to the similar period in 2019. In brief, the partial lockdown in the Mexico-city showed a negative growth mainly due to abnormality in daily mean temperature, which affected vegetation, water content, and LST, which could not influence the urban environment towards positive development.

## Greater Mumbai city (Fig 3)

The Greater Mumbai city was under total lockdown on March 25, 2020, which was subsequently extended till May 31, 2020. During March-April, the city received negligible rainfall (during lockdown: 0.08 mm in 2019 & 0.17 mm in 2020) [35], with similar trends during pre-lockdown. This substantiates no influence of scanty rainfall on the urban greenness due to hot

and dry weather during pre- and lockdown, compared to the similar period in 2019 (Table 1, Fig 3, and S1 Table in S1 File). Similar weather conditions and due to total restrictions on outdoor activities during the lockdown, the pre-lockdown deficit in urban greenness was completely nullified, which increased during post-lockdown mainly due to heavy monsoon rainfall.

There was surplus water content in water bodies during pre-lockdown (Fig 3, S1 Table in S1 File) mainly in the Sanjay Gandhi national park water reservoir, and also in lockdown period in Ulhas and Kalundre River. Later, during the post-lockdown, the increment in water content was reduced to zero, since heavy southwest monsoon rainfall nullified all variations that occurred due to lockdown. A significant reduction in LST during the pre-lockdown compared to that in 2019 (Fig 3, S3 Table in S1 File) was due to overcast conditions throughout the day, and the temperature and rainfall values were similar. Similar weather conditions during the lockdown led to lower LST than the comparable period in 2019. However, a slight increase in air temperature led to an increment in LST during 2020 post-lockdown [39, 40].

During pre-lockdown, active light radiance was lesser than the 2019 similar phase due to reduction in industrial/ transportation activities due to Holi–a Hindu festival of colors. Lower NTL during lockdown is attributed to the total lockdown (Fig 3, S1 Table in S1 File). Moreover, a significant increment in the inactive lights during post-lockdown was found between Navi Mumbai and Thane, Chembur, and Trombay. The total lockdown was the main cause for lower active NTL in the greater Mumbai. During the lockdown, the LST and NTL, which signifies outdoor anthropogenic activities and anthropogenic heat flux at a surface level, were highly benefitted due to total lockdown restrictions; however, water crisis could become a major issue during the lockdown in this megacity.

## Sao Paulo city (Fig 4)

Sao Paulo implemented a partial lockdown on March 23, 2020, during which transportation, pharmacies, and grocery shops were operational, while population movement was restricted to 75–80% [54]; however, by April, only 50% of the population was under partial lockdown restrictions [43, 44]. Urban greenness was reduced during pre-lockdown due to the low rainfall during 2020 (14-day mean: 0.679 cm) compared to a similar period in 2019 (14-day mean: 11.54 mm) (Fig 4, S1 Table in S1 File). The increase in the greenness during lockdown compared to that in 2019 was due to lockdown restrictions even though negligible rainfall was reported and a degree difference in air temperature (similar date last year) was observed(Fig 4); similar positive effects continued during post-lockdown (August 2020), which is the driest month in Sao Paulo. The trend in NDWI during pre-and lockdown phases (Fig 4, S3 Table in S1 File) suggests that the water content in the city was influenced by similar factors as that affected urban greenness. However, post lockdown reduction was due to a deficit in the water content at Repressa Billings.

The LST was lower during pre-lockdown due to a slight reduction in air temperature compared to the similar period in 2019 (Fig 4, S1 Table in S1 File). In contrast, LST during lockdown saw additional reduction (1.33%) primarily due to partial lockdown restrictions, which led to low anthropogenic heat flux. As a result, increment in lockdown phase radiance was mainly observed along BR-116. No restrictions on transportation during the partial lockdown phase led to higher NTL along the highway BR-116 passing through the city and in areas such as the south-central zone, which is in constant development with residential neighborhoods and Liberdade (where floating lanterns are hung on streets). It proves that residential areas were illuminated more during the lockdown phase compared to normal days.

## Toronto city (Fig 5)

Pre-lockdown was imposed in the early week of March when the land was covered by snow, which makes NDVI values lower than actual [45]. The air temperature was nearly 0˚C during lockdown (April), which led to the low NDVI values compared to the 2019 similar period when the air temperature was 6˚–15˚C on the date of data acquisition [39, 40]. This also suggests that the lockdown phase in 2020 had higher snow accumulation than the similar period in 2019.

The spectral reflectance from the snow-covered area is inversely proportional to the water content in the snow [46], which suggests that during pre-lockdown in 2020, the extent of snow-covered land was more than in 2019, which makes NDWI values lower than actual, which explains for the lower water content during pre-lockdown (Fig 5, S1 Table in S1 File). The lockdown was during springs when a higher amount of snowmelt occurs, which caused high water content. However, these trends were disrupted due to higher water consumption in household activities [41, 42] which yielded low water content during the lockdown. Further, increment in water content was observed during post-lockdown phases as water demand for household consumption eased.

The LST was mainly influenced by prevailing weather. According to MODIS imageries, the LST of the Old Toronto region was found to be exceptionally high during pre-lockdown. However, the LST dropped during the lockdown phase compared to a similar period in 2019 (Fig 5, S1 Table in S1 File) due to meltwater on the land surface. A similar trend was observed during the post-lockdown phase.

A significant reduction in active lights during the pre-lockdown phase compared to a similar period in 2019 was possibly due to people staying indoors hearing the news of contagious COVID-19 spreading across the world. As a developed nation, soft precautions might have been taken, which affected economic activities, and hence the active lights. The NTL remained lower than the 2019 similar phase during lockdown due to little vehicular mobility and reduced economic activities (Fig 5, S1 Table in S1 File). Later, the NTL increased during post-lockdown compared to a similar period in 2019 due to the resumption of economic activities. In brief, the urban environment variables of the Toronto city were primarily influenced by local weather more than the lockdown measures.

## Interpretation of expert response to the environment-related SDGs

A total of 104 responses of domain experts from different continents were recorded. Based on the response score, 14 targets from 6 different SDGs were selected, which were found to be directly influencing the urban environment due to lockdown restrictions (Fig 6). The response rate for option "Agree" was between 43.3% (target 15.1) and 92.3% (target 11.6). While "Disagree" response rate was between 3.8% (target 7.3) and 29.8% (target 15.1). An additional response option, "Can't say" was specially added in this survey, and interestingly, the response rate for these options was found to be between 2.9% (target 11.6) and 31.7% (target 15.2), better than the option "Disagree" which certainly made the survey less biased.

The response rate for cumulative SDG targets under the urban vegetation category was 60.78% (Agree), 20.38% (Disagree), and 18.82% (Can't say). Cumulative targets regarding urban water content received 61.525% (Agree), 18.75% (Disagree), and 19.725% (Can't say). Land Surface Temperature target received 70.2% (Agree), 12.5% (Disagree), and 18.82 (Can't say). Urban active lights targets received 82.06% (Agree), 7.68% (Disagree), and 10.22% (Can't say) responses. While urban air quality target obtained 92.3% (Agree), 4.8% (Disagree), and 2.9% (Can't say) responses. In brief, in terms of the cumulative positive impact of lockdown

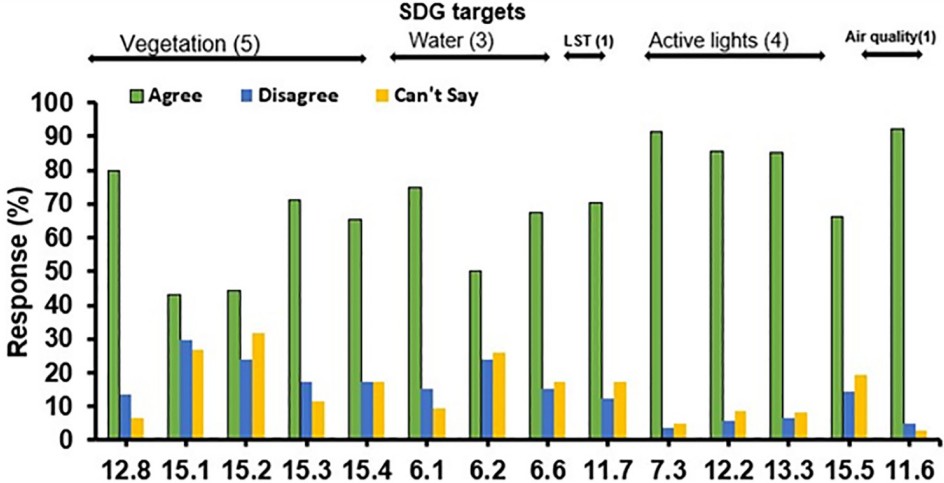

**Fig 6. The percentage share of expert responses for each UN-SDG target under different urban environmental categories.**

on total SDG targets, 70% of total SDG targets responded in affirmative, while the remaining 14.6% got equal shares of "Disagree" and "Can't say" responses.

In pursuit of economic development, degradation of the urban natural environment increases by many folds in these humungous cities (alpha cites are a typical example), and this trend continues for several decades, which has become one of the impending global crises (to create sustainable cities). Restoration and preservation of degraded natural ecosystems should be achieved by increasing international cooperation to combat climate change [47]. However, regardless of their intensity, human activities have benefits and repercussions on the natural environment in a specific manner. The COVID-19 induced lockdown restrictions created a short-term environment revival in the modern era by interrupting the global business-as-usual scenario.

Generally, the urban natural environment in the metropolis is influenced by local topography, weather (rainfall and temperature), and anthropogenic activities that causes many environmental problems [48–50] including urban heat island, pollution, etc. Local weather, in turn, gets heavily influenced by anthropogenic activities (human-induced climate change), and static local topography is rapidly modified by the anthropogeomorphic process. Thus, weather and human activities and development through industrialization, transportation, etc., play a primary role in influencing the urban natural environment. Therefore, implementation of lockdown restriction possesses a similar magnitude to affect the urban environment with different intensities.

The spatial distribution of NDWI, LST, NDVI, and NTL for pre-, during and post-lockdown phases in 2019 and 2020 for the five alpha cities is shown in Figs 1 to 5. Though no significant changes were observed during pre and during lockdown periods, a substantial increase (20%) in NDWI was deciphered in Seputeh and Bandar Tun Razak, Segambut, Titiwangsa (Fig 1). LST showed a decrease in LST in pre-lockdown 2020 in Seputeh, Kepong, western Lembah Pantai. During the lockdown, 30% reduction in the LST all over the city. Significant decreases in LST (>40%) were found during post-lockdown phase almost over the entire city. NDVI during pre-lockdown increased by 30% at many pockets in Segambut and other districts on the west Kuala Lumpur. A substantial increase in vegetation is observed

during lockdown and post-lockdown. NTL increased in pre- and during lockdown, but decreased (15%) in Segabut district (high commercial town) in the post-lockdown phase. In brief, there was a marginal change in NDWI confined to the south, a decrease in LST, an increase in vegetation, and active lights increased during and post-lockdown period (Fig 1).

For Mexico city, the NDWI decreased in the pre-lockdown in small pockets in the southernmost Milpa Alta and increase in eastern Xochimilco districts, with more than 33% in the post-lockdown phase in the south. The LST showed decrease by 7˚C in the eastern regions from 2019 data for pre- and during lockdown. The vegetation increased by 60% in the southern Milpa Alta region and decrease 30% in eastern Iztapalapa during post- and lockdown period (Fig 2). Active lights showed no appreciable change across the phases in the urbanized areas. No appreciable changes were detected in NDWI for greater Mumbai across the three phases (Fig 3). LST increased at the airport area, but decreased at the Sanjay Gandhi park during pre- and post-lockdown. Significant changes in NTL were not widespread.

At least a 30% reduction in the NDWI for the river basins was observed for Sao Paulo city across the three phases of lockdown. Urbanized areas in Leste, Nordeste, and Sudeste depicted higher changes in the LST, with a 3–5˚C increase for all the three phases. We encountered decrease in vegetation (25%) in the urban areas, possibly due to an increase in the LST during pre-lockdown 2020, increase in NDVI by 35% during lockdown (Fig 4). We found an increase in radiance from the surface (NTL) in pre- and during the lockdown phases by 40% in urban areas. For Toronto city, we found an increase in NDWI, possibly due to more melt in pre- and post-lockdown 2020. An overall increase by 2˚C in pre-lockdown 2020 with a blob in LST (4˚C) in the south portion of Old Toronto was observed. We found Toronto city warmer during lockdown 2020 and 6˚C cooler in post-lockdown 2020. Overall greenness increased by 50% during pre-lockdown 2020, which decreased in Old Toronto/ East York and increased east Scarborough by 30% during lockdown 2020 (Fig 5). A significant positive trend in greenness was found for post-lockdown phase 2020. NTL increased/decreased by 10% in pre-lockdown/ during lockdown 2020; an increase in NTL was detected in the post-lockdown phase.

## Lockdown link to environment parameters-related UN-SDG targets

Positive NDVI values in Kuala Lumpur, Mexico city, greater Mumbai, and Sao Paulo during and post-lockdown phases strongly suggests the positive influence of COVID-19 related restrictions on environment-related three SDG and their five targets. The urban environment parameters and the links to UN-SDGs as observed from the present stauy are given in Table 2.

**The overall impact of lockdown upon Alpha city environment.** The COVID-19 induced lockdown restrictions irrespective of their nature had positively affected at least more than one environmental variable in each of the cities. In addition, local climatic conditions were equally responsible and sometimes even dominated lockdown restriction influence (for example, Toronto); however, the findings show that Greater Mumbai and Toronto, which were under a total lockdown, both had observed positive influence on cumulative urban environment (although Toronto was affected mainly by local weather conditions than lockdown) (Table 3). While in other cities (Mexico City, Sao Paulo) where partial lockdown was implemented, cumulative lockdown effects were found to be in deficit for a similar period in 2019, mainly due to partial restrictions on transportation and shopping activities (Table 3). The only exception was Kuala Lumpur which observed surplus growth while having partial lockdown because the restrictions were only partial during the Muslim festival of Ramadan, which later was tightened. It implies that total lockdown measures put restrictions on all kinds of economic activities and population movement, which significantly reduce anthropogenic involvement in urban environmental processes and result in significant positive growth. Otherwise, any

**Table 2. Lockdown and its link to environmental parameters and UN-SDG targets.**

|  | NDVI(Vegetation) | NDWI(Water) | LST(Temperature) | NTL(Power/Energy) | Air Quality |
|---|---|---|---|---|---|
| **UN-SDG Links** | It is evident from the results that the pandemic and consequent lockdown restrictions have increased awareness about sustainability & local environment protection (SDG–12, target 12.8) (Fig 6) Nevertheless, there is a change in perception across the world that a similar type of pandemic is not far away if humans do not reduce overexploitation of natural resources leading to catastrophic events as the disintegration of polar ice sheets (particularly west Antarctic and Greenland), melting of mountain glaciers due to rising temperatures, strong cyclones, heat waves, etc., could introduce more deadly dormant virus [51] to inhabitable regions. | Water shortage is a chronic problem in the majority of areas across the globe. Although it is proven that lockdown measures led to an increase in water demands in urban households [41, 42], however, at the same time, lockdown measures had assisted in increasing the source of potable water (SDG–6, target 6.1) by cleaning up freshwater streams and other water bodies as a result of restrictions on industrial operations, transportation, and population movement, as most of these water bodies were not in an ideal state for human consumption before lockdown. COVID-19 lockdown has assisted in understanding that if left undisturbed, polluted water bodies can also clean themselves within a shorter time (SDG–15 target 15.1) which could be revolutionary in formulating future water policies. | COVID-19 lockdown restrictions led to an increment in urban vegetation and reduced LST (greater Mumbai and Sao Paulo). This trend is highly suitable to make green and sustainable public spaces more accessible in urban areas. Higher LST, due to heat island effect, in urban area inconvenient for city dwellers [52–54] especially for children, women, and elderly persons who use green public spaces (SDG–7, target 11.7) (Fig 6). | COVID-induced lockdown led to restrictions on economic activities; industrial operations, transportation, and other commercial activities within the city had stopped [48]. The night-time lights are the indicator of human activities, and due to lockdown restrictions in greater Mumbai and Kuala Lumpur, it was significant reduced primarily in commercial areas and along the major highways, railways, and airports due to less traffic which also led to a significant reduction in energy consumption (SDG–7, target 7.3). This also indicates the reduction in material footprints (SDG–12, target 12.2) due to reduced unnecessary shopping activities and limited supplies of consumable items. | COVID-19 lockdown-induced reduction in air pollution such $PM_{10}$, $PM_{2.5}$, CO, $NO_2$, $O_3$, and $SO_2$ and consequent improvement in air quality are one of the most evident and well-documented phenomena of positive effects, not only in the large metropolis of Kuala Lumpur [55] (Suhaimi)], Mexico City [37], Mumbai [56, 57], Sao Paulo [17, 58] and Toronto [18] but also in small towns such as Barcelona [59] (Tobías)], City of Brescia [60] (Cameletti)] and industrial cities [17, 48] primarily in highly polluted South Asian countries. |
| **Lockdown measures** | Lockdown restrictions led to minimal or no movement of population, transportation, and curtailed economic activities. These measures have also positively assisted in the restoration of degraded land and vegetation (SDG–15, target 15.1 & 15.3), reduction in deforestation (SDG–15, target 15.2), improvement in mountain vegetation (SDG–15, target 15.4). | The contagious nature of COVID-19, along with lockdown restrictions, compelled people to stay indoors, which led to an increment in hygiene levels (SDG–6, target 6.2). |  | Furthermore, minimal economic activities led to reduced industrial waste (smoke, untreated chemical discharge, etc.), which helped natural habitats such as wetlands, forests, and near urban areas to breathe (SDG–15, target 1). |  |
| **Net Effects** | Cumulatively, COVID-19 lockdown has contributed significantly towards actions to reduce degradation of natural habitat, thus fulfilling SDG-15, target 15.5 as well. | In addition, an increment in available water content in Sao Paulo urban area was also observed during lockdown (SDG–6, target 6.6) (Fig 6). | Lockdown restrictions assisted in bringing LST down. | The reduction in NTL due to COVID-19 lockdown also promoted work from the home culture, which could save unnecessary daily commute using high emission transportation services, energy consumption at offices, railway stations, etc., thereby reducing per capita energy consumption (SDG–13, target 13.3) (Fig 6). | This complements the SDG–11, target–11.6 (Fig 6). |

**Table 3. Impact of COVID-19 lockdown restriction on Alpha cities relative to similar period in 2019.**

| City | Nature of COVID-19 lockdown restrictions | Growth in all urban environmental variables (%) during lockdown phase relative to a similar period in 2019 |
|---|---|---|
| Kuala Lumpur | Partial | +21.55 |
| Mexico City | Partial | -10.56 |
| Greater Mumbai | Total | +13.76 |
| Sao Paulo | Partial | -1.23 |
| Toronto | Total | +13.92 |

activity that remained unrestricted would interfere in this process. Hence, the total lockdown restriction was found to be more effective in influencing urban environmental variables than the partial lockdown.

## Conclusion

The present study tried to explore all the possibilities of influence of COVID-19 lockdown restrictions on the urban natural environment and UN-SDG during these challenging times. Despite there being some limitation of the investigations such as use of low-resolution remote sensing data, MODIS (due to cloud cover on high-resolution acquisition dates) and absence of field validation of (due to spatial extensiveness of the study and restrictions on domestic and international travel), the study still holds novelty to investigate positive effects of lockdown on different urban environmental variables (except air quality), which were not extensively assessed earlier and subsequent positive influence upon already jeopardized UN-SDGs.

The growth in the urban environmental variables during lockdown phase 2020 relative to a similar period in 2019 varied from 13.92% for Toronto to 13.76% for greater Mumbai to 21.55% for Kuala Lumpur; it dropped to −10.56% for Mexico and −1.23% for Sao Paulo city. The total lockdown was more effective in revitalizing the urban environment than partial lockdown. Our results also indicated that Greater Mumbai and Toronto, which were under a total lockdown, had observed positive influence on cumulative urban environment. While in other cities (Mexico City, Sao Paulo) where partial lockdown was implemented, cumulative lockdown effects were found to be in deficit for a similar period in 2019, mainly due to partial restrictions on transportation and shopping activities. The only exception was Kuala Lumpur which observed surplus growth while having partial lockdown because the restrictions were only partial during the festival of Ramadan.

The COVID-19 induced lockdown has undoubtedly impacted the global economy affecting almost all the nations, but some advantages of the lockdown measures are visible in the environmental parameters resulting in reduced pollution and enhanced vegetation in many cities. The present study on the five major alpha cities across the world supports these changes that have affected both adversely and positively. The study brought up interesting findings which would be helpful for future researches. "Research like this provides evidence for promoting teleworking (work-from-home or home-office) policies and investing in public transportation and electrification" [61]. Additionally, the findings provide a different perspective to perceive lockdown measures as the a buzz of negativity with a silver lining of social connectedness [62] to achieve some of the jeopardized UN-SDGs.

## Supporting information

**S1 File.**
(DOCX)

## Acknowledgments

The Ph.D. candidate Ritwik Nigam acknowledges the financial support provided by the University Grant Commission (UGC), Govt. of India, New Delhi, to conduct this research. AJL thanks Director, NCPOR for research facilities.

## Statement

We emphasize that all methods were carried out in accordance with relevant guidelines and regulations.

We assert that all relevant experimental protocols were approved by the ethics committee of Goa University.

We declare that the relevant informed consent was obtained from all subjects involved in this study.

## Author Contributions

**Conceptualization:** Gaurav Tripathi, Alvarinho J. Luis, Eric Vaz, Achala Shakya, Mahender Kotha.

**Data curation:** Ritwik Nigam, Achala Shakya.

**Formal analysis:** Ritwik Nigam, Tannu Priya, Alvarinho J. Luis, Shashikant Kumar.

**Investigation:** Ritwik Nigam, Gaurav Tripathi, Tannu Priya, Shashikant Kumar, Achala Shakya, Mahender Kotha.

**Methodology:** Ritwik Nigam, Tannu Priya, Eric Vaz, Shashikant Kumar, Achala Shakya.

**Resources:** Tannu Priya.

**Software:** Ritwik Nigam, Gaurav Tripathi, Alvarinho J. Luis, Shashikant Kumar, Mahender Kotha.

**Supervision:** Eric Vaz, Bruno Damásio, Mahender Kotha.

**Validation:** Eric Vaz, Bruno Damásio, Mahender Kotha.

**Visualization:** Ritwik Nigam, Gaurav Tripathi, Mahender Kotha.

**Writing – original draft:** Ritwik Nigam, Tannu Priya, Alvarinho J. Luis, Eric Vaz, Shashikant Kumar, Achala Shakya, Bruno Damásio, Mahender Kotha.

**Writing – review & editing:** Eric Vaz, Bruno Damásio, Mahender Kotha.

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
