## [Decision Letter · Decision Letter 0]

19 Jan 2022

PONE-D-21-39297Did Covid-19 lockdown positively affect the urban environment and UN- Sustainable Development Goals?PLOS ONE

Dear Dr. Damásio,

Thank you for submitting your manuscript to PLOS ONE. After careful consideration, we feel that it has merit but does not fully meet PLOS ONE’s publication criteria as it currently stands. Therefore, we invite you to submit a revised version of the manuscript that addresses the points raised during the review process.

We look forward to receiving your revised manuscript.

Kind regards,

Bailang Yu

Academic Editor

PLOS ONE

Journal Requirements:

3. We note that Figure 2 in your submission contain [map/satellite] images which may be copyrighted. All PLOS content is published under the Creative Commons Attribution License (CC BY 4.0), which means that the manuscript, images, and Supporting Information files will be freely available online, and any third party is permitted to access, download, copy, distribute, and use these materials in any way, even commercially, with proper attribution. For these reasons, we cannot publish previously copyrighted maps or satellite images created using proprietary data, such as Google software (Google Maps, Street View, and Earth). For more information, see our copyright guidelines: http://journals.plos.org/plosone/s/licenses-and-copyright.

Additional Editor Comments:

I agree with the reviewers that the lockdown policy affects our environment but not all indicators. The authors should provide more information about the relationships between the policy and the variations of the indicators. 

Reviewers' comments:

Reviewer's Responses to Questions

**Comments to the Author**

1. Is the manuscript technically sound, and do the data support the conclusions?

Reviewer #1: Partly

Reviewer #2: Yes

Reviewer #3: Partly

2. Has the statistical analysis been performed appropriately and rigorously? 

Reviewer #1: Yes

Reviewer #2: No

Reviewer #3: Yes

3. Have the authors made all data underlying the findings in their manuscript fully available?

Reviewer #1: No

Reviewer #2: Yes

Reviewer #3: Yes

4. Is the manuscript presented in an intelligible fashion and written in standard English?

Reviewer #1: Yes

Reviewer #2: No

Reviewer #3: Yes

5. Review Comments to the Author

Reviewer #1: Title: Did Covid-19 lockdown positively affect the urban environment and UN-Sustainable Development Goals?

Manuscript Number: PONE-D-21-39297

Journal name: PLOS ONE

It is my pleasure to review the manuscript for the esteemed journal. In this manuscript, the authors investigated the effects of the lockdown impacts with regards to a set of urban environmental parameters (greenness, land surface temperature, night- time light, and energy consumption) in five alpha cities (Kuala Lumpur, Mexico, greater Mumbai, Sao Paulo, Toronto) about the environment-related UN Sustainable Development Goals (SDGs) using an extensive questionnaire-based survey of expert opinions. The work presented is relevant to the Journal's field. The manuscript has got some potential. I would like to congratulate the author for a considerable amount of work that they have done. Especially, the author reported that the total lockdown was more effective in revitalizing the urban environment than partial lockdown. This manuscript has provided a new case to more comprehensive understanding th effects of lockdown on the environment in cities. However, the manuscript needs further improved before to be accepted for publication. The reviewer has listed some specific comments that might be helpful of the author to further enhance the quality of the manuscript. Please consider the particular comments listed below.

Comment 1: Abstract. Generally it should further underscore the scientific value added of your paper in your abstract. In addition, it seems that there are too many descriptions of the background and methods of the research, but the description of the results is too simplistic.

Comment 2: sections of Introduction. The novelty of this manuscript should be further justified by highlighting main contributions to the existing literature. This could be clearly presented in the introduction section. Please consider citing following papers which are related to your research: (i) https://doi.org/10.1016/j.envres.2021.111990; (ii) https://doi.org/10.1016/j.jclepro.2021.127897; (iii) https://doi.org/10.3390/ijerph18116053; (iv) https://doi.org/10.1016/j.spc.2021.04.024; (v) https://doi.org/10.1016/j.jclepro.2021.126265. on one hand, the current introduction seems simple. One the other hand, there has already been a large amount of literatures discussing this topic. There is a need to better elaborate the contribution of the work to the existing literature.

Comment 3: sections of methodology. The section is well-structured and well-written. The detailed description of the method is impressive. However, it would be better to highlight your improvement of the method and your innovation in methods.

Comment 4: section of Result interpretation. The section is also well-written. However, it seems that there is no in-depth discussion in both Empirical Results and future research. It would be better to further discuss what your findings are different from the past works.

Comment 5: sections of conclusion. Please make sure your conclusions' section underscore the scientific value added of your paper, and/or the applicability of your findings/results, as indicated previously. Basically, you should enhance your contributions, limitations, underscore the scientific value added of your paper, and/or the applicability of your findings/results and future study in this session.

Comment 6: There are still some occasional grammar errors through the revised manuscript especially the article ''the'', ''a'' and ''an'' is missing in many places, please make a spellchecking in addition to these minor issues.

Comment 7: References. Please check the references in the text and the list; You should update the reference.

Reviewer #2: This paper explores impacts of lockdowns against COVID-19 epidemic on urban environment. The paper adopted remotely sensed indices and compared environmental changes before and after the lockdown periods. Although environmental impacts of this pandemic have been reported in numerous previous studies, this paper provides some useful information to assess and understand environmental influences of this pandemic. However, there are still many shortcomings should be improved before this paper is in consideration of publication.

Major concerns:

1. The lockdown policy indeed affects our environment, but not all indicators. In terms of specific indicators, it is encouraged to express clear routines from lockdown-related restrictions to environmental consequences. For example, the author adopted NDWI (water index) that was used to extract the area of water bodies. However, those lockdown-related restrictions did not affect the area of water bodies in a short time (several months or one year).

2. Why did the authors choose four indictors (NDWI, LST, NDVI, and NTL)? Are there any interrelationships among these four indicators? Or would you like to tell a combined story based on these four indicators? If the answer is not sure, maybe you should focus on one indicator in depth, such as the night-time lights.

3. More explanations on links between four indicators and UN-SDGs are encouraged in the Introduction section. In my opinion, there four indicators are weakly correlated to UN-SDGs.

4. Sub-sections and sub-titles of this paper are not arranged in a regular manner. I am confused that there is a “Results Interpretation” section and one more “Result” section. And also, other sub-titles are not concise and easy-reading.

Minors:

1. Why authors chose these five cities? Are they all alpha cities? Are they representative samples among alpha cities?

2. The results of experiments are compressed in only one figure (Fig. 2). Readers are hard to find differences of environments before and after lockdowns. The figure should be separated into several figures or you should pay attention to only one or two indicators.

Reviewer #3: This paper explored the positive effects of COVID-19 lockdown restrictions on environment-related SDGs. Some of the urban environment variables extracted from satellite images showed variation pre-, during, and post-lockdown. I found this topic interesting and would provide some valuable information. However, there are some flaws with the results that I hope the authors continue to work on methodologies to improve the results. Please see my comments below.

1.In the introduction section, the authors state that “Originated in Wuhan city of China in early December 2019, the Coronavirus disease that causes COVID-19 has global scale transmission.” As far as I know, the geographic origin of the COVID-19 virus has not yet been found, so the authors’ statement is not rigorous.

2.What are the exact date ranges of “pre-”, “post”- and “during” lockdown phases of each city? What are the corresponding dates of remote sensing images? For partial lockdown, what is the specific area of the lockdown? A more detailed formulation of the above questions could make the analysis more convincing.

3.In the result interpretation part, the authors describe the changes of urban environment variables. I suggest to represent them in the form of a table to make it clearer.

4.The reliability of the analysis results needs to be further improved. Firstly, the authors mainly use mean value to analyze the changes of urban environment variables in each city. I suggest that the authors further locate the areas with obvious changes, and analyze the reasons for the changes in combination with the land use situation. Secondly, according to the authors' analysis, most of the changes are caused by local weather rather than the lockdown measures. The current analysis can hardly support the conclusion that lockdown has an considerable impact on environment-related SDGs. More quantitative studies will need to be conducted with more related data.

5.To study the impact of partial lockdown on the city environment, I suggest that the authors further analyze the differences of environment variables within and outside the lockdown area.

6. PLOS authors have the option to publish the peer review history of their article (what does this mean?). If published, this will include your full peer review and any attached files.

Reviewer #1: No

Reviewer #2: No

Reviewer #3: No

---

## [Author Response · Author response to Decision Letter 0]

22 Apr 2022

The authors express their sincere thanks to the editor for his valuable time and for giving us an opportunity to respond to the comments of the reviewers which was very insightful and helped us in further improving the quality of manuscript to enable us to publish in this esteemed journal of repute.

All the comments of the 3 reviewers have been addressed and responses to the same are given just below their comments, respectively (with separate colored text) for easy understanding. 

Reviewer #1: Title: Did Covid-19 lockdown positively affect the urban environment and UN-Sustainable Development Goals?

It is my pleasure to review the manuscript for the esteemed journal. In this manuscript, the authors investigated the effects of the lockdown impacts with regards to a set of urban environmental parameters (greenness, land surface temperature, night- time light, and energy consumption) in five alpha cities (Kuala Lumpur, Mexico, greater Mumbai, Sao Paulo, Toronto) about the environment-related UN Sustainable Development Goals (SDGs) using an extensive questionnaire-based survey of expert opinions. The work presented is relevant to the Journal's field. The manuscript has got some potential. I would like to congratulate the author for a considerable amount of work that they have done. Especially, the author reported that the total lockdown was more effective in revitalizing the urban environment than partial lockdown. This manuscript has provided a new case to more comprehensive understanding the effects of lockdown on the environment in cities. However, the manuscript needs further improved before to be accepted for publication. The reviewer has listed some specific comments that might be helpful of the author to further enhance the quality of the manuscript. Please consider the particular comments listed below.

Authors thank the reviewer for the insightful comments on the previous draft version that helped us to improve further the quality of the manuscript.

Comment 1: Abstract. Generally it should further underscore the scientific value added of your paper in your abstract. In addition, it seems that there are too many descriptions of the background and methods of the research, but the description of the results is too simplistic.

Response: The abstract is re-written and now embodies the findings of the study.

Comment 2: sections of Introduction. The novelty of this manuscript should be further justified by highlighting main contributions to the existing literature. This could be clearly presented in the introduction section. Please consider citing following papers which are related to your research: (i) https://doi.org/10.1016/j.envres.2021.111990; (ii) https://doi.org/10.1016/j.jclepro.2021.127897; (iii) https://doi.org/10.3390/ijerph18116053; (iv) https://doi.org/10.1016/j.spc.2021.04.024; (v) https://doi.org/10.1016/j.jclepro.2021.126265. on one hand, the current introduction seems simple. One the other hand, there has already been a large amount of literatures discussing this topic. There is a need to better elaborate the contribution of the work to the existing literature.

Response: The suggested references relevant to COVID-19 scenario have been referred to in the introduction. (at Line No.49 and 62 references included at line nos. 559-574)

Comment 3: sections of methodology. The section is well-structured and well-written. The detailed description of the method is impressive. However, it would be better to highlight your improvement of the method and your innovation in methods.

Response: The methods adopted in this work are innovative in the following way.

1. The satellite-based parameters which are normally used to access climate change are used to decipher the post-lockdown changes in the alpha cities in different climatic zones

2. We have used survey to explore and analyze the linkages to environment-related SDGs.

Comment 4: section of Result interpretation. The section is also well-written. However, it seems that there is no in-depth discussion in both Empirical Results and future research. It would be better to further discuss what your findings are different from the past works.

Response: To our knowledge, there are no works just prior to the COVID-19 period and so comparison is not possible. 

Comment 5: sections of conclusion. Please make sure your conclusions' section underscore the scientific value added of your paper, and/or the applicability of your findings/results, as indicated previously. Basically, you should enhance your contributions, limitations, underscore the scientific value added of your paper, and/or the applicability of your findings/results and future study in this session.

Response: Modified (at Line No.498-516)

Comment 6: There are still some occasional grammar errors through the revised manuscript especially the article ''the'', ''a'' and ''an'' is missing in many places, please make a spellchecking in addition to these minor issues.

Response: Checked and rectified

Comment 7: References. Please check the references in the text and the list; You should update the reference.

Response: Redundant references have removed.

Reviewer #2: This paper explores impacts of lockdowns against COVID-19 epidemic on urban environment. The paper adopted remotely sensed indices and compared environmental changes before and after the lockdown periods. Although environmental impacts of this pandemic have been reported in numerous previous studies, this paper provides some useful information to assess and understand environmental influences of this pandemic. However, there are still many shortcomings should be improved before this paper is in consideration of publication.

Authors thank the reviewer for the insightful comments on the previous draft version that helped us to improve further the quality of the manuscript.

Major concerns:

1. The lockdown policy indeed affects our environment, but not all indicators. In terms of specific indicators, it is encouraged to express clear routines from lockdown-related restrictions to environmental consequences. For example, the author adopted NDWI (water index) that was used to extract the area of water bodies. However, those lockdown-related restrictions did not affect the area of water bodies in a short time (several months or one year).

Response: Though no immediate direct effect of lock down on water bodies is possible in general, the variable (NDWI) is linked to NDVI and NDBI, was specifically chosen to understand the effect on water bodies (if any) in urban regions by comparing with NDVI (vegetation), as Vegetation (NDVI) is more closely associated with NDWI. 

2. Why did the authors choose four indictors (NDWI, LST, NDVI, and NTL)? Are there any interrelationships among these four indicators? Or would you like to tell a combined story based on these four indicators? If the answer is not sure, maybe you should focus on one indicator in depth, such as the night-time lights.

Response: Four indicators are chosen to compare and bring-out any indirect/subtle changes and since there were some specific observations showing spatio-temporal variations of these variables, it is thought that a combined account of four indicators is included in the study which will surely provide an impetus to future studies by others to take up. 

3. More explanations on links between four indicators and UN-SDGs are encouraged in the Introduction section. In my opinion, there four indicators are weakly correlated to UN-SDGs.

Response: This paper attempts for the first time attempted tried to correlate the four environmental parameters (NDVI, NDWI, LST and NTL) with corresponding UN-SDGs. More the links between four indicators and UN-SDGs was included separately and not as part of introduction to restrict the length of introduction.

4. Sub-sections and sub-titles of this paper are not arranged in a regular manner. I am confused that there is a “Results Interpretation” section and one more “Result” section. And also, other sub-titles are not concise and easy-reading.

Response: Sub-sections and sub-titles of this paper are now re-arranged in a regular manner and the results part is rewritten as the descriptive part is now arranged in Tabular Format (Table 1 and Table 2) (at Line No.781 and 782)

Minors:

1. Why authors chose these five cities? Are they all alpha cities? Are they representative samples among alpha cities?

Response: The availability of data, economic background, and their representativeness in different continents were some characteristics for selecting these 5 random alpha cities (Kuala Lumpur, Mexico, greater Mumbai, Sao Paulo, and Toronto).

2. The results of experiments are compressed in only one figure (Fig. 2). Readers are hard to find differences of environments before and after lockdowns. The figure should be separated into several figures or you should pay attention to only one or two indicators.

Response: The results expressed in one single figure (Fig.2) are now separated into 5 separate figures (city-wise) as Figs.1 to 5 (at Line Nos. 269, 271, 287, 289, 316, 318, 344, 346, 368, 370) 

Reviewer #3: This paper explored the positive effects of COVID-19 lockdown restrictions on environment-related SDGs. Some of the urban environment variables extracted from satellite images showed variation pre-, during, and post-lockdown. I found this topic interesting and would provide some valuable information. However, there are some flaws with the results that I hope the authors continue to work on methodologies to improve the results. Please see my comments below.

Authors thank the reviewer for the insightful comments on the previous draft version that helped us to improve further the quality of the manuscript.

1.In the introduction section, the authors state that “Originated in Wuhan city of China in early December 2019, the Coronavirus disease that causes COVID-19 has global scale transmission.” As far as I know, the geographic origin of the COVID-19 virus has not yet been found, so the authors’ statement is not rigorous.

Response: Removed

2.What are the exact date ranges of “pre-”, “post”- and “during” lockdown phases of each city? What are the corresponding dates of remote sensing images? For partial lockdown, what is the specific area of the lockdown? A more detailed formulation of the above questions could make the analysis more convincing.

Response: The details are given in Table-1 under supplementary material

3.In the result interpretation part, the authors describe the changes of urban environment variables. I suggest to represent them in the form of a table to make it clearer.

Response: The details of variation in each of the parameters is now given in Table-1 & 2 (at Line No.781 and 782)

4.The reliability of the analysis results needs to be further improved. Firstly, the authors mainly use mean value to analyze the changes of urban environment variables in each city. I suggest that the authors further locate the areas with obvious changes, and analyze the reasons for the changes in combination with the land use situation. Secondly, according to the authors' analysis, most of the changes are caused by local weather rather than the lockdown measures. The current analysis can hardly support the conclusion that lockdown has an considerable impact on environment-related SDGs. More quantitative studies will need to be conducted with more related data.

Response: The mean value is used as a routine to analyze and interpret the changes in urban environment variables. The location specific observations are illustrated in the Figs 1 to 5. The due to non-availability of land use data for the corresponding periods, the location specific changes could not be combined with land use data.

5.To study the impact of partial lockdown on the city environment, I suggest that the authors further analyze the differences of environment variables within and outside the lockdown area.

Response: Although this was in our agenda, was not attempted in the present study as this study was mainly focused on to observe the spatio-temporal changes.

6. PLOS authors have the option to publish the peer review history of their article (what does this mean?). If published, this will include your full peer review and any attached files.

Response: Authors have no objection to publish the peer review history of the article.

---

## [Decision Letter · Decision Letter 1]

20 Jun 2022

PONE-D-21-39297R1Did Covid-19 lockdown positively affect the urban environment and UN- Sustainable Development Goals?PLOS ONE

Dear Dr. Damásio,

Thank you for submitting your manuscript to PLOS ONE. After careful consideration, we feel that it has merit but does not fully meet PLOS ONE’s publication criteria as it currently stands. Therefore, we invite you to submit a revised version of the manuscript that addresses the points raised during the review process.

We look forward to receiving your revised manuscript.

Kind regards,

Bailang Yu

Academic Editor

PLOS ONE

Additional Editor Comments:

I agree with the reviewer 3's comments that the authors should recheck the reliability of the results.

Reviewers' comments:

Reviewer's Responses to Questions

**Comments to the Author**

1. If the authors have adequately addressed your comments raised in a previous round of review and you feel that this manuscript is now acceptable for publication, you may indicate that here to bypass the “Comments to the Author” section, enter your conflict of interest statement in the “Confidential to Editor” section, and submit your "Accept" recommendation.

Reviewer #1: (No Response)

Reviewer #2: All comments have been addressed

Reviewer #3: (No Response)

2. Is the manuscript technically sound, and do the data support the conclusions?

Reviewer #1: (No Response)

Reviewer #2: Yes

Reviewer #3: Partly

3. Has the statistical analysis been performed appropriately and rigorously? 

Reviewer #1: (No Response)

Reviewer #2: Yes

Reviewer #3: Yes

4. Have the authors made all data underlying the findings in their manuscript fully available?

Reviewer #1: (No Response)

Reviewer #2: Yes

Reviewer #3: Yes

5. Is the manuscript presented in an intelligible fashion and written in standard English?

Reviewer #1: (No Response)

Reviewer #2: Yes

Reviewer #3: Yes

6. Review Comments to the Author

Reviewer #1: The authors have incorporated comments from the first round of review. My concerns from my previous review have been addressed. I would recommend the paper to be accepted for publication. Thank you!

Reviewer #2: The author has revised the paper in response to my questions. There are 2 more issues that need further revision.

First, the abstract of the paper only describes some results and does not give a conclusion. Author(s) did not respond to the question in the title.

Second, the pictures in this paper do not meet the quality requirements for publication. As shown in Figure 1-5, it is recommended to cancel the labeling of latitude and longitude, and be sure to increase the font size in the figure.

Reviewer #3: The authors have modified the paper to address most of my comments. But I still insist that the reliability of the results needs to be further improved. It would be better to locate the areas with obvious changes, and analyze the reasons for the changes in combination with the land use situation. If no land use data is available, it is also possible to refer to Google Image data for analysis. In addition, I still think that most of the changes are caused by local weather rather than the lockdown measures according to the authors' analysis. It would be better to conduct more quantitative studies.

7. PLOS authors have the option to publish the peer review history of their article (what does this mean?). If published, this will include your full peer review and any attached files.

Reviewer #1: No

Reviewer #2: No

Reviewer #3: No

---

## [Author Response · Author response to Decision Letter 1]

18 Jul 2022

Response to Reviewers

Tile of Manuscript: Did Covid-19 lockdown positively affect the urban environment and UN- Sustainable Development Goals? 

Manuscript Number: PONE-D-21-39297; Journal name: PLOS ONE

The authors express their sincere thanks to the editor for his valuable time and for giving us an opportunity to respond to the comments of the reviewers which was very insightful and helped us in further improving the quality of manuscript to enable us to publish in this esteemed journal of repute.

All the comments of the 3 reviewers have been addressed and responses to the same are given just below their comments, respectively (with separate colored text) for easy understanding. 

Reviewer #1: The authors have incorporated comments from the first round of review. My concerns from my previous review have been addressed. I would recommend the paper to be accepted for publication. 

Authors thank the reviewer for the insightful comments on the previous draft version that helped us to improve further the quality of the manuscript.

Reviewer #2: The author has revised the paper in response to my questions. There are 2 more issues that need further revision.

First, the abstract of the paper only describes some results and does not give a conclusion. Author(s) did not respond to the question in the title.

Response: The authors have added the specific conclusion with regard to fulfilling UN-SDG 

Second, the pictures in this paper do not meet the quality requirements for publication. As shown in Figure 1-5, it is recommended to cancel the labeling of latitude and longitude, and be sure to increase the font size in the figure.

Response: Thanks for the suggestion with regard to the Figures 1 to 5. The figures have been revised as suggested by removing the coordinates labeling and also the increased the font size of the text for better clarity.

Reviewer #3: The authors have modified the paper to address most of my comments. But I still insist that the reliability of the results needs to be further improved. It would be better to locate the areas with obvious changes, and analyze the reasons for the changes in combination with the land use situation. If no land use data is available, it is also possible to refer to Google Image data for analysis. In addition, I still think that most of the changes are caused by local weather rather than the lockdown measures according to the authors' analysis. It would be better to conduct more quantitative studies.

Response: While we appreciate the reviewer’s remarks on further improvements in the contextual assemblage of land-use change, we are confident that the reviewer recognizes that understanding the local impacts of land-use change would likely be a study in its own right and not one of the analyses of the regional effects we propose. Furthermore, an empirical analysis of land-use data and proceeding with local classification for multiple regions would be a project in its own right. Issues related to satellite imagery, particularly cloud coverage’s impact on a precise outcome in the classification, would make this a relatively coarse endeavor.

Further the Manuscript is checked for any additional grammatical/spelling mistakes.

Authors express their sincere thanks to the editor and the reviewers for their valuable time in review process with their insightful comments/observations on the manuscript that have truly helped the authors to present their study in a more appealing way. 

Thanking you,

Bruno Damásio

(on behalf of all authors)

---

## [Decision Letter · Decision Letter 2]

1 Sep 2022

Did Covid-19 lockdown positively affect the urban environment and UN- Sustainable Development Goals?

PONE-D-21-39297R2

Dear Dr. Damásio,

We’re pleased to inform you that your manuscript has been judged scientifically suitable for publication and will be formally accepted for publication once it meets all outstanding technical requirements.

Kind regards,

Bailang Yu

Academic Editor

PLOS ONE

Additional Editor Comments (optional):

Reviewers' comments:

Reviewer's Responses to Questions

**Comments to the Author**

1. If the authors have adequately addressed your comments raised in a previous round of review and you feel that this manuscript is now acceptable for publication, you may indicate that here to bypass the “Comments to the Author” section, enter your conflict of interest statement in the “Confidential to Editor” section, and submit your "Accept" recommendation.

Reviewer #2: All comments have been addressed

Reviewer #3: All comments have been addressed

2. Is the manuscript technically sound, and do the data support the conclusions?

Reviewer #2: (No Response)

Reviewer #3: Yes

3. Has the statistical analysis been performed appropriately and rigorously? 

Reviewer #2: Yes

Reviewer #3: Yes

4. Have the authors made all data underlying the findings in their manuscript fully available?

Reviewer #2: Yes

Reviewer #3: Yes

5. Is the manuscript presented in an intelligible fashion and written in standard English?

Reviewer #2: Yes

Reviewer #3: Yes

6. Review Comments to the Author

Reviewer #2: The author(s) have addressed all my concerns. I would like to suggest to accept this paper for publication in PloS One.

The following two newly published papers are for your reference, which are closely related to nighttime lights changes during the COVID-19 epidemic.

----Rowe, F., C. Robinson & N. Patias (2022) Sensing global changes in local patterns of energy consumption in cities during the early stages of the COVID-19 pandemic. Cities, 129, 103808.

----Xu, G., T. Xiu, X. Li, X. Liang & L. Jiao (2021) Lockdown Induced Night-Time Light Dynamics during the COVID-19 Epidemic in Global Megacities. International Journal of Applied Earth Observation and Geoinformation, 102, 1-10.

Reviewer #3: (No Response)

7. PLOS authors have the option to publish the peer review history of their article (what does this mean?). If published, this will include your full peer review and any attached files.

Reviewer #2: No

Reviewer #3: No

---

## [Editor Report · Acceptance letter]

12 Sep 2022

PONE-D-21-39297R2 

Did Covid 19 lockdown positively affect the urban environment and UN- Sustainable Development Goals? 

Dear Dr. Damásio:

I'm pleased to inform you that your manuscript has been deemed suitable for publication in PLOS ONE. Congratulations! Your manuscript is now with our production department. 

Kind regards, 

on behalf of

Dr. Bailang Yu 

Academic Editor

PLOS ONE